# Descriptions of Entropy with Fractal Dynamics and Their Applications to the Flow Pressure of Centrifugal Compressor

**DOI:** 10.3390/e21030266

**Published:** 2019-03-08

**Authors:** Yan Liu, Dongxiao Ding, Kai Ma, Kuan Gao

**Affiliations:** School of Mechanical Engineering, Northwestern Polytechnical University, Xi’an 710072, China

**Keywords:** entropy, fractal dynamics, centrifugal compressor, surge, hurst exponent, multi-fractal spectrum

## Abstract

In this study, some important intrinsic dynamics have been captured after analyzing the relationships between the dynamic pressure at an outlet of centrifugal compressor and fractal characteristics, which is one of powerful descriptions in entropy to measure the disorder or complexity in the nonlinear dynamic system. In particular, the fractal dynamics of dynamic pressure of the flow is studied, as the centrifugal compressor is in surge state, resulting in the dynamic pressure of flow and becoming a serious disorder and complex. First, the dynamic pressure at outlet of a centrifugal compressor with 800 kW is tested and then obtained by controlling the opening of the anti-surge valve at the outlet, and both the stable state and surge are initially tested and analyzed. Subsequently, the fractal dynamics is introduced to study the intrinsic dynamics of dynamic pressure under various working conditions, in order to identify surge, which is one typical flow instability in centrifugal compressor. Following fractal dynamics, the Hurst exponent, autocorrelation functions, and variance in measure theories of entropy are studied to obtain the mono-fractal characteristics of the centrifugal compressor. Further, the multi-fractal spectrums are investigated in some detail, and their physical meanings are consequently explained. At last, the statistical reliability of multi-fractal spectrum by modifying the original data has been studied. The results show that a distinct relationship between the dynamic pressure and fractal characteristics exists, including mono-fractal and multi-fractal, and such fractal dynamics are intrinsic. As the centrifugal compressor is working under normal condition, its autocorrelation function curve demonstrates apparent stochastic characteristics, and its Hurst exponent and variance are lower. However, its autocorrelation function curve demonstrates an apparent heavy tail distribution, and its Hurst exponent and variance are higher, as it is working in an unstable condition, namely, surge. In addition, the results show that the multi-fractal spectrum parameters are closely related to the dynamic pressure. With the state of centrifugal compressor being changed from stable to unstable states, some multi-fractal spectrum parameters Δ*α*, Δ*f*(*α*), *α_max,_* and *f*(*α_min_*) become larger, but *α_min_* in the multi-fractal spectrum show the opposite trend, and consistent properties are graphically shown for the randomly shuffled data. As a conclusion, the proposed method, as one measure method for entropy, can be used to feasibly identify the incipient surge of a centrifugal compressor and design its surge controller.

## 1. Introduction

As one type of flow instability, surge will occur if the flow of the centrifugal compressor is reduced to a critical value [1]. Surge, which is an intrinsic characteristic of the centrifugal compressor, can lead to lower aerodynamic performance and flow-induced vibration, giving rise to the instability of aero-elasticity or fatigue of structure as surge becomes violent. In particular, the dynamic pressure of flow becomes a serious disorder and complex as the centrifugal compressor is in transition from stable state to surge, and the routine methods are difficult to capture and describe the complex characteristics.

In recent years, scientists have paid more attention to studying surge. It is believed that the internal factor, which may induce the surge, is the rotation stall in passage and the separation of flow around the blades, and the external factor is the high pressure gas that is stored in the pipes. However, some researchers think that the rotation stall and surge are two different flow instability phenomena, and the stall is the prelude to surge for the centrifugal compressor [2]. In terms of control strategies, Moghaddam used a decoupled sliding-mode neural network method to control surge [3]. Shu Lin proposed a fuzzy control method for axial compressor that does not need a surge line [4]. Liao and Chen utilized nonlinear wavelet analysis to deal with the stall in different blade scales [5], and then the wavelet processing technology was applied to the prediction of compressor instability. With further research, scientists found that it is difficult to accurately model the compressor due to the complicated internal flow. Accordingly, researchers began to use the neural network to approximate the nonlinear function in order to predict surge, instead of modeling surge [6]. Yet, most of the researches still focus on the theoretical study or the numerical model, and the surge active control method, such as neural network, which would not appear to be a very practical method [7]. 

As studied in this field, it is found that flow pressures from inlet, outlet, and impeller behave as nonlinear dynamics as the centrifugal compressor is working in the surge state. That is, the time series of flow pressure becomes singular, even chaotic, and some intrinsic property in time series should exist. Fortunately, entropy could be introduced to describe and measure such singular phenomenon. Specifically, fractal dynamics, as one of powerful descriptions in entropy, could be applied to describing and measuring such singular phenomenon. Indeed, entropy is an extension of a known concept in information entropy for time-dependent processes, which is used to characterize dynamical systems between regular, chaotic, and purely random evolution [8]. Additionally, fractal dimensionality and largest Lyapunov exponent, as the measures of the entropy and state functions, have been intensively and exhaustively used in the last 30 years in the study of nonlinear dynamics [9,10,11]. Based on such motivation, fractal dynamics are used to describe the characteristic of the flow pressure for the centrifugal compressor. 

In this paper, some fundamental theories in entropy, which are related to mono-fractal, Hurst exponent and multi-fractal spectrum etc., are introduced and developed firstly to describe the fractal dynamics of flow pressure and its intrinsic properties. Subsequently, the relationships between fractal dynamics and the real time series of outlet flow pressure for centrifugal compressor are further studied. Finally, some results are obtained, and the feasibility of the method to identify surge is accurately verified, and the results will provide an effective way to predict the surge.

## 2. Fundamental Theory

### 2.1. Mono-Fractal and Related Parameters

#### 2.1.1. Mono-Fractal 

Mono-fractal, which is also called self-similarity, means that the characteristics of certain processes are identical or similar from different scales of dimension or time. Autocorrelation function, Hurst exponent, and variance are used to describe the mono-fractal characteristics of time series. Indeed, they are some measure methods for entropy. 

(1) Auto-correlation function

The auto-correlation function, as a function of times or time lags, could show the correlation between the values of a random process at different times.

If the expected value *E*(*x_i_*) and variance var(*x_i_*) of a random process *x_i_* exist, then the auto-correlation function *r*(*k*) can be presented as the following formula,
(1)$$r(k)=\frac{E[(x_{i+k}-E(x_{i}))(x_{i}-E(x_{i}))]}{\mathrm{var}{\left(x_{i}\right)}^{2}},k\in I$$
where *k* is time lag. For a stationary process, the auto-correlation function rapidly attenuates to zero, along with the increase of *k*. On the other hand, it slowly attenuates for an unstable process [12,13].

(2) Heavy-Tailed distribution

In probability theory, heavy-tailed distributions are probability distributions that have heavier tails than the exponential distribution [14]. 

For a random process *x_i_* with distribution function *F*(*X*) = *P*{*X* < *x*}, if the following formula exists, then the process will show a heavy-tailed distribution, namely,
(2)$$1\;-\;F(x)\;=\;P\{X>\;x\}\;\sim \frac{1}{x^{a}},\mathrm{when}\;x\;\to \;\infty ,\;a\;>\;0$$

The large variance and slowly decreasing auto-correlation function curve of a random process make it have heavy-tail characteristics, which is closely related to the bursting phenomena.

(3) Hurst exponent and Rescaled Range Analysis

The Hurst exponent related to the auto-correlations is used as a parameter to measure the long-term memory of time series, and it is usually calculated by Rescaled Range (*R/S*) Analysis. 

To estimate the Hurst exponent, a time series $X_{\alpha }(\alpha =1,2,3,\ldots N)$ is divided into a number of shorter time series *X_k_* (*k* = 1,2, …, *n* and *n* ≥ 3), and length *n* = *N*, *N*/2, *N*/4, … The average rescaled range is then calculated for each value of *n*. For time series *X_k_*, the rescaled range is calculated, as follows [15]:

1. Calculate the mean value;
(3)$$m=\frac{1}{n}\sum_{k=1}^{n}X_{k}$$

2. Create a mean-adjusted series;
(4)$$Y_{t}=X_{t}-m\;\;(t=1,2,\ldots n)$$

3. Calculate the cumulative deviate series *Z*;
(5)$$Z_{t}=\sum_{i=1}^{t}Y_{i}\;\;(t=1,2,\ldots n)$$

4. Define range *R*(*n*) as the difference between the maximum and the minimum;
(6)$$R(n)=\underset{1\leq t\leq n}{\max}(Z_{t})-\underset{1\leq t\leq n}{\min}(Z_{t})$$

5. Compute the standard deviation *S*(*n*);
(7)$$S(n)={\left[\frac{1}{n}\sum_{k=1}^{n}{\left(X_{k}-m\right)}^{2}\right]}^{\frac{1}{2}}$$

6. Calculate the rescaled range *R*(*n*)/*S*(*n*);
(8)$$R(n)/S(n)=Cn^{H}$$
where *C* is a constant. Transform the formula and the Hurst exponent is estimated by fitting the power law
(9)log[R(n)/S(n)]=Hlogn+logC

The value of Hurst is obtained by plotting log[*R*(*n*)/*S*(*n*)] as a function of log *n* and fitting a straight line by the lease square-method. Following the definition, the slope of the straight line is the Hurst exponent.

##### 2.1.2. Hurst Exponent and Dynamic Pressure

The Hurst exponent (*H*) could quantify the relative tendency of a time series either to strongly regress to the mean or to cluster in a direction. Based on the theory of statistics, the available range of *H* is from 0 to 1. 

*H* in the range 0–0.5 indicates the time series with long-term switching between high and low values in adjacent pairs, meaning that a high outlet pressure value will probably be followed by a low value for centrifugal compressor, and the switch trend between the high and low pressure will last for a long time in the future.

*H* = 0.5 indicates that the time series with the law of random walk, known as a stochastic or random process, which describes a path that consists of a succession of random steps on some mathematical space, such as the integers. In this case, the correlation between the pressures of adjacent moments is weak. 

*H* in the range 0.5–1 indicates that the time series with long-term positive autocorrelation, which means that a high outlet dynamic pressure in the series will be probably followed by another high pressure. In other words, the pressures of adjacent moments have great relevance to each other. In this case, the higher *H* indicates a higher degree of self-similarity.

Consequently, the Hurst exponent can be regarded as a quantitative method for describing the dynamic pressure, for characterizing its burst characteristics, and for predicting the initial surge of the compressor.

### 2.2. Multi-Fractal Spectrum and Related Parameters

A multi-fractal system is a generalization of a fractal system in which a single exponent (the fractal dimension) is not enough to describe its dynamics. Instead, a continuous spectrum of exponents (the so-called singularity spectrum) is needed [16]. Multi-fractal cannot be decomposed to separate fractals. However, by means of tools of multi-fractal analysis, we have information regarding fractal composition, including the complexity of signal. In a sense, there exists a scale property in the dynamic pressure at the outlet of centrifugal compressor with non-uniform fractal.

The properties of long-range correlation in the large time scale can be studied by self-similarity. However, multi-fractal is a time-dependent and local singular behavior, and it can be used in the study of bursting in real time series with a small time scale [17]. Normally, two main variables, namely, Hŏlder exponent, multi-fractal spectrum can describe the characteristics of multi-fractal in the time series of dynamic pressure, which are introduced in the following.

#### 2.2.1. Multi-Fractal and Variables of Multi-Fractal Spectrums

In a sense, a multi-fractal is thought of as one kind of decomposition of a fractal object, and the decomposed components will have their own fractal dimension. Assume that there exists a stochastic process {*x*(*t*), *t* ≥ 0}, and it is split into several sub-intervals with a length *δ* for each interval. Subsequently, an exponent for measuring singularity of *x*(*t*) at time *t*_0_ can be defined by the following expression,

(10)$$\alpha (t_{0})=\underset{\delta \to 0}{\lim}\frac{\ln(x(t))}{\ln\delta }$$

The singularity exponent *α*(*t*) is also referred to as the Hŏlder exponent, which is used to describe the fractal dimension in the fractal geometric theory, that is, it can measure the probability of the growing of the small interval [18]. If the *α*(*t*) of a stochastic process *x*(*t*) is time-dependent, that is, the scale property is related to time, then the time series or process possess multi-fractal properties. Moreover, if *α*(*t*) is a constant, then the singularities of the time series can just be expressed in term of one global singularity exponent, that is, the series possess a mono-fractal property. In such a situation, the singularity exponent *α*(*t*) is referred to the Hurst exponent.

In order to analyze the stochastic process, it can be split into several sub-intervals. If every sub-intervals is “narrow” enough, the *α*(*t*) in the sub-interval can be assumed as a constant, and then multi-fractal spectrum *f*(*α*) can be used to analyze the fractal property of the stochastic process. Indeed, the multi-fractal spectrum is the probability of same *α*(*t*) in a series composed of an infinite number of *α*(*t*). Hence, the value of the *f*(*α*) is within [0,1].

#### 2.2.2. Relationships between Variables of Multi-Fractal Spectrums 

The multi-fractal spectrum *f*(*α*) can be used to quantitatively describe the distribution of probability in the evolution of the system, and the statistic method is used to study the distribution of probability. The relationships between the variables of multi-fractal spectrums can be obtained, as the following. In this paper, let *P*_i_(*ε*) denote the change rate of normalized dynamic pressure with time scale *ε*, at *i*th interval, namely,
(11)$$P_{i}(\varepsilon )=I_{i}/\sum I_{i}$$where *I_i_* is the total dynamic pressure at outlet with time scale *ε*, at *i*th interval, and then $\sum_{i=1}^{N}P_{i}=1$.

It is clear that the dynamic pressure becomes higher as *P_i_*(*ε*) increases. Let *α* be the singularity exponent in *i*th interval. If the time series of dynamic export pressure in this period shows multi-fractal properties, then the following power law relationship is satisfied:(12)$$P_{i}(\varepsilon )\propto \varepsilon^{\alpha }$$

As stated above, *α* could describe the singularity in each interval. It is clear that *P_i_* will become large as *α* increases, since *ε* < 1, and vice versa. 

A function *χ_q_*(*ε*), namely, the partition function for the multi-fractal system, can be defined by the following,
(13)$$\chi_{q}(\varepsilon )=\sum_{i=1}^{N}P_{i}{\left(\varepsilon \right)}^{q}$$where, *q* is the weighting factor that is used to decompose a fractal structure into several sub-levels. Subsequently, *χ_q_*(*ε*) can measure the non-uniform of *P_i_*. If there exist a multi-fractal in the time series of dynamic pressure, then there exists a power-type relationship between the distribution function and *ε*, and the structure function can be defined as

(14)$$\tau (q)=\frac{\ln\chi_{q}(\varepsilon )}{\ln\varepsilon }$$

In Equation (14), *τ*(*q*) is the structure function, which can be obtained by applying least square fit to the straight line part of the curve described by ln*χ_q_*(*ε*)～ln*ε*. 

The mono-fractal can be considered as a special case of multi-fractal in a sense, that is, as the Hŏlder exponent *α*(*t*) is a constant at any time for a system, the multi-fractal properties can reduce into mono-fractal properties. Following the Legendre Transformation [18], there exists a structure function
(15)τ(q)=qH(q)−1
where, *H* is Hurst parameter [18], that is, *τ*(*q*) is a linear function of *H*.

If the process is multi-fractal, then there exist some relationships between *α*, *f*(*α*), and *τ*(*q*), as follows,

(16)f(α)=q⋅α−τ(q)

(17)$$\alpha =\frac{d\tau (q)}{dq}$$

#### 2.2.3. Application of Multi-Fractal Spectrum to Dynamic Pressure

As previously mentioned, the singularity exponent *α*(*t*) is used as a measure of localization characteristics of time series, so it can describe the different dynamic behaviors of dynamic pressure for centrifugal compressors [12].

If *α*(*t*) is a constant, then the singularity of the time series on all of the time scales can be described by only one exponent, which is the typical characteristic of mono-fractal. If *α*(*t*) is the function of time *t*, that is, the characteristic of its scale is the function of time, then the time series has a multi-fractal property. In comparison with the mono-fractal, the introduction of multi-fractal extends the understanding of the time scales, and the scale that is relevant to time could describe the non-regular behaviors in the local time interval [19].

In the expression of *α*(*t*), *α_min_* refers to the maximum probability of *P_i_*(*ε*) and *α_max_* means minimum probability. Hence, *α*(*t*) can be introduced to capture the complicated behaviors of outlet dynamic pressure and Δ*α* can be defined as the multi-fractal spectrum width as in Equation (18), which could describe the property of non-uniform of the probability for the entire fractal. Accordingly, the characteristic, which is related with the different level and local area, can be described in detail. 

Δ*α* = *α_max_* − *α_min_*(18)

The dimension of the subset of the series that is characterized by *α* is denoted by multi-fractal spectrum *f*(*α*) and its structure has a close relationship with the complex dynamics of the system [18]. Once surge of centrifugal compressor occurs, the multi-fractal spectrum will obviously shift along the *α* axis, and the span of curve *f*(*α*) will become large rapidly. As a summary, there will be a possibility of the surge of centrifugal compressor, as the width of multi-fractal spectrum is large in a time interval.

In addition, the multi-fractal spectrum parameters, such as *α*_max_ and *f*(*α*_max_), represent the properties of the minimum subset of the probability, and *α*_min_ and *f*(*α*_min_) represent the properties of the maximum subset of the probability. Hence, multi-fractal spectrum can be considered to be a measure for the complexity, non-regularity, and non-uniform properties of the fractal structure of the real time series. Consequently, define the multi-fractal parameter Δ*f*(*α*) to explain the proportion of dynamic pressure in wave crest or trough in all time, as in Equation (19) [20],
Δ*f*(*α*) = *f*(*α_min_*) − *f*(*α_max_*)(19)

As Δ*f*(*α*) > 0, the shape of the multi-fractal spectrum curves is right-hook, meaning that the proportion of dynamic pressure in wave crest in a time interval is large, implying surge will occur easily. More, as Δ*f*(*α*) < 0, the shape of the multi-fractal spectrum curves is left-hook, the dynamic pressure in a time interval is at a low value and the working state of the centrifugal compressor is normal.

## 3. Data Acquisition and Spectrum Analysis of Dynamic Pressure

### 3.1. Data Acquisition System

Figure 1 shows the diagram of data acquisition system. Three parts comprise the acquisition system: the static acquisition system, dynamic acquisition system, and the monitoring and diagnosing system. The virtual instrument made of NI is used as the static acquisition system, whose main function is to monitor the temperature, mass flow, the inlet and out static pressure, and the rotor velocity of direct-current machine. The dynamic acquisition system could test the dynamic pressure at the outlet of centrifugal compressor. The monitoring and diagnosing system mainly monitor the shaft vibration and axis displacement and the later diagnosis analysis. 

As shown in Figure 1, the centrifugal compressor is driven by direct-current machine. At first, the air enters the working pipeline through an air filter and chamber. Subsequently, the compressor pumps the air. Finally, the compressed air is billowed into the atmosphere through exhaust muffler by controlling the two electric anti-surge valves with diameters of 250 mm and 100 mm. The experiment is carried out by controlling the opening-degree of the two electric anti-surge valves, and the centrifugal compressor transits from the stable state to surge, with the opening-degree decreasing gradually. The dynamic pressure testing experiment lasts for 635.5 s from stable state to surge, and then to stable state again for centrifugal compressor.

In the paper, the data from 180 s to 330 s, totally 150 s, are used, including the stable state and the surges of centrifugal compressor transiting. Accordingly, the 180 s is used as the starting time in our study. The parameters of the acquisition system are shown in Table 1.

### 3.2. Frequency Spectrum of Dynamic Pressure

The time series of the dynamic pressure at the outlet of centrifugal compressor is shown in Figure 2, and the frequency spectrums of different stages are presented in Figure 3. As shown in Figure 2, it is clear that the centrifugal compressor is in the stable state with the anti-surge valve fully open, within the first 80 s. In this situation, the amplitude of frequency spectrum for dynamic pressure is less and the fluctuation of dynamic pressure is faster. With the decrease of the opening of anti-surge valve, the amplitude of frequency spectrum begins to gradually increase, and the fluctuation slows down after 80 s. Afterwards, the surge occurs at 95 s nearly.

As shown in Figure 3, the dynamic pressure has a broad spectrum in the stable state, meaning that the frequency is incoherent. As the fluctuation deepens, the higher amplitude of the frequency spectrum becomes concentrate to the lower frequency band, and when the centrifugal compressor is in surge state, as in Figure 3d. This phenomenon means that the dynamic pressure series in the stable state is random. However, the spectrum analysis indicates that the dynamic pressure series become periodic to some degree as surge occurs.

## 4. Mono-Fractal Characteristics of Dynamic Pressure

Autocorrelation, Hurst exponent, and variance are used to describe the mono-fractal characteristics of the dynamic export pressure. With the data of 0–10 s, 80–90 s, 90–100 s, and 100–110 s, the similarity of dynamic pressure at the outlet of the centrifugal compressor system is analyzed in this section. 

### 4.1. Autocorrelation Characteristics

Four sets of autocorrelation curves are shown in Figure 4, and each set includes 10 curves for continuous time periods. In the diagrams, *k* is the lag and the maximum value is 400. From Figure 4, the 10 autocorrelation curves for first 0–10 s is rapidly attenuated, indicating the weak correlation. With a decrease of the anti-surge valve opening and the slow attenuation of the curves, the highly heavy-tailed feature is displayed.

### 4.2. Hurst Exponent and Variance of Dynamic Pressure 

The Hurst exponent and the variance of dynamic pressure series are shown in Figure 5 and Figure 6, respectively. From Figure 5, it shows that all of the Hurst exponents are in the range of 0.5–1 in every condition, which means that the dynamic pressure series is self-correlated. Furthermore, when a surge occurs, the Hurst exponents significantly increase, showing a highly self-similarity trend. As the opening of anti-surge valve decreases, the variance shown in Figure 6 increases, indicating the heavy-tailed feature. The result in Figure 6 is similar to the Hurst curve in Figure 5.

In Figure 5, there is a sudden change for Hurst exponent during 86–96 s, increasing first and then decreasing. Hence, a further study on initial surge is necessary. Table 2 shows the Hurst exponent and the variance of dynamic pressure series in one second as a unit to analyze the stage of initial surge.

From Figure 5 and Table 2, it can be seen that the time periods of 92–96 s are in the stage of initial surge. There are unstable pressures and larger Hurst exponent in this stage. Accordingly, the pressure series in period of 92–96 s show the most obvious long correlation, and the burst takes place easily in this period. 

After 96 s, the system enters the surge state. The Hurst exponent decreases slightly, but it is still much higher than in the stable state. At this state, the long correlation characteristic, the relevance between the data, and the mutation are all slightly weakened. Therefore, the analysis of the Hurst exponent is useful in the determination of the changing trend of dynamic pressure.

## 5. Nonlinear Behavior of Structure Function for Dynamic Pressure

A nonlinear behavior of *τ*(*q*) can be considered as a manifestation of multi-scaling. In Figure 7, in order to better visualize the scaling character of the data, the relationship between structure function *τ*(*q*) and weighting factor *q* for the dynamic pressure at the outlet is analyzed. Four time series in different stages are used, which are stable state (0–4 s), transition (90–94 s), and surge state (100–104 s, 140–144 s), and every stage is divided into four groups. In Equation (15), *τ*(*q*) is a linear function of *H*. Mono-fractal signals (*H*(*q*) = const) are associated with a linear plot *τ*(*q*), while the multi-fractal ones possess the nonlinear spectra in *q*. The nonlinearity of *τ*(*q*) is confined to the central range of *q*’s around *q*= 0. While for larger values of |*q*|, the behavior of *τ*(*q*) is almost linear due to the finite size of the sample [21]. In Figure 7, the nonlinearity of *τ*(*q*)’s is much weaker for the data of stable state than those of the surge state, and the strongest multifractality is attributes of the data of transition. Our calculations indicate that the time series of the dynamic pressure at the outlet can be of the multi-fractal nature.

## 6. Relationships between Multi-Fractal Spectrum and Fluctuation of Dynamic Pressure

Using the presented method, multi-fractal spectra are used to decompose a fractal structure into several sub-levels. Figure 8 shows the multi-fractal spectrum graphs for dynamic export pressure of different stages in 0–10 s, 80–85 s, 85–90 s, 90–95 s, 95–100 s, and 100–110s, and every stage is divided into some groups. Figure 9 shows temporal variation of the multi-fractal spectrum parameters. All of the parameters of multi-fractal spectrum are calculated by using the data of the dynamic pressure in one second as a unit. 

Theoretically, the absolute value of *q* should be infinite for the calculations of the multi-fractal spectrum. In practices, the multi-fractal spectrum is not changed with the increase of the absolute value of *q*, when the absolute value of *q* increases to a certain value. It will also cause the overflow errors for the computer if the absolute value of *q* is large to some degree. Accordingly, in the system, we choose *q*∈(−80, 80) in the stable state and surge stage, and *q*∈(−20, 20) in the transition stage for calculating multi-fractal spectrums in order to ensure the integrity of the spectrum and escape the overflow errors of computer. 

From Figure 8 and Figure 9, some conclusions can be drawn, as follows:

The patterns of multi-fractal spectrum that are shown in Figure 8 are typically humped, reflecting the general characteristics of multi-fractal spectrum. The shapes of the most multi-fractal spectrums show a right-hook. Obviously, all of the Δ*α* of the multi-fractal spectra increase with the average flow, as the curve shows in Figure 8 and Figure 9, that is, the pressure distribution becomes non-uniform and the corresponding fluctuation sharply increases with the average pressure increase.

In 0–85 s, the values of Δ*α* are approaching zero. All of the values of *α_min_* are large, while the values of *α_max_* are small. It indicates that the working states of the centrifugal compressor system are basically similar and less likely to burst in the periods. 

In 85–96 s, the values of Δ*α*, Δ*f*(*α*), *α_max_* and *f*(*α_min_*) dramatically increase, while the values of *α_min_* reduce rapidly. In this state, Δ*α*, Δ*f*(*α*) and *α_max_* and *f*(*α_min_*) reach the maximum, and *α_min_* nearly reach the minimum. High changing parameter means that the dynamic pressure dramatically changes, implying that surge occurs easily in this stage.

In 96–150 s, the compressor is in a deep surge and the values of Δ*α* are large. The larger Δ*α* indicates that the proportion of the dynamic pressure is larger in a wave crest in a time interval. In the stage, all of the parameters begin to oscillate or fluctuate, indicating the unstable of the system. The parameters in this stage are very different from those in stable state. 

From the parameters of multi-fractal spectrum, Δ*α*, *α_max_* and *α_min_* can show the state of the system clearly, while the other parameters could not imply the state obviously. Accordingly, the future work is to study the statistical characteristics of the relevant parameters. 

## 7. The Statistical Reliability of Multi-Fractal Spectrum for Dynamic Pressure

In present paper, in order to test the statistical reliability of the multi-fractal spectrum, we modified the data of dynamic pressure by adding randomly shuffled data. We choose the ten percent of the maximum absolute data of the dynamic pressure as the maximum absolute data of the randomly shuffled data. Figure 10 shows the comparison of the original and randomly shuffled dynamic pressure in multi-fractal spectrum parameters, which shows *f*(α) spectrums of six time intervals as 1–2 s, 82–83 s, 90–91 s, 92–93 s, 100–101 s, and 101–102 s. Figure 11 shows Δα and Δ*f*(α) of the randomly shuffled data. From Figure 10 and Figure 11, the shapes of the most multi-fractal spectrums both show a right-hook in the original data and the randomly shuffled data. At the stable state, the width of *f*(α) spectrum implies being slightly wider for randomly shuffled data. Moreover, Δα and Δ*f*(α) both show graphically consistent properties, except for little errors. 

## 8. Conclusions

In order to accurately identify surge, the fractal dynamics of dynamic pressure at outlet of centrifugal compressor is studied to obtain the intrinsic property as one powerful measure method for entropy. The results show that the dynamic pressure at the outlet of the centrifugal compressor in stable state slightly fluctuates and exhibits stochastic characters. When the system enters surge, the pressure series show periodicity to a certain level, complex fine levels of the time series, heavy-tail, and self-similar characters. After careful studying, the results of the mono-fractal and multi-fractal are highly consistent with multi-fractal describing more detailed change of system state. That is, the parameters of the Hurst exponent and multi-fractal spectrum can clearly show the transition process of the system, namely, from stable state to surge. Our next work is using our results to predict the initial surge of centrifugal compressor in order to prevent surge.

## Figures and Tables

**Figure 1 entropy-21-00266-f001:**
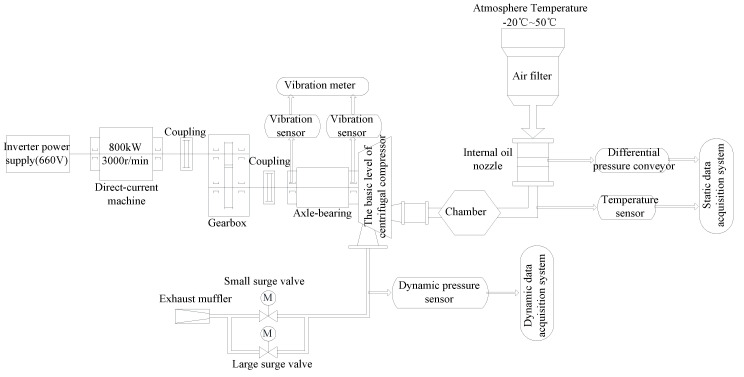
Acquisition system.

**Figure 2 entropy-21-00266-f002:**
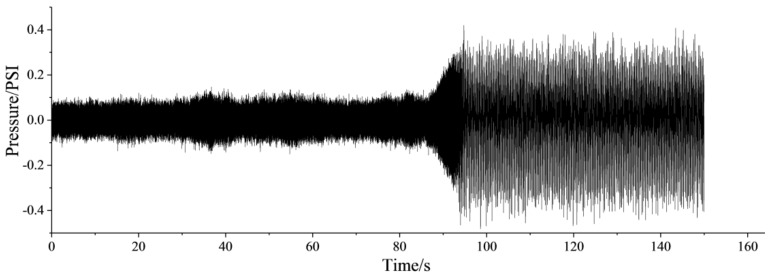
Time series of dynamic pressure.

**Figure 3 entropy-21-00266-f003:**
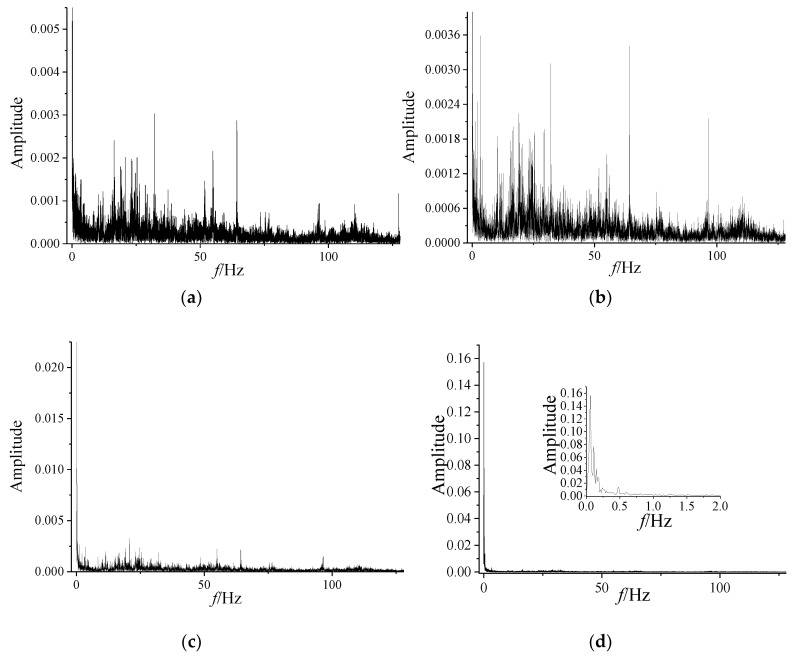
Frequency spectrum of dynamic pressure. (**a**) 0–10 s; (**b**) 80–90 s; (**c**) 90–100 s; (**d**) 100–110 s.

**Figure 4 entropy-21-00266-f004:**
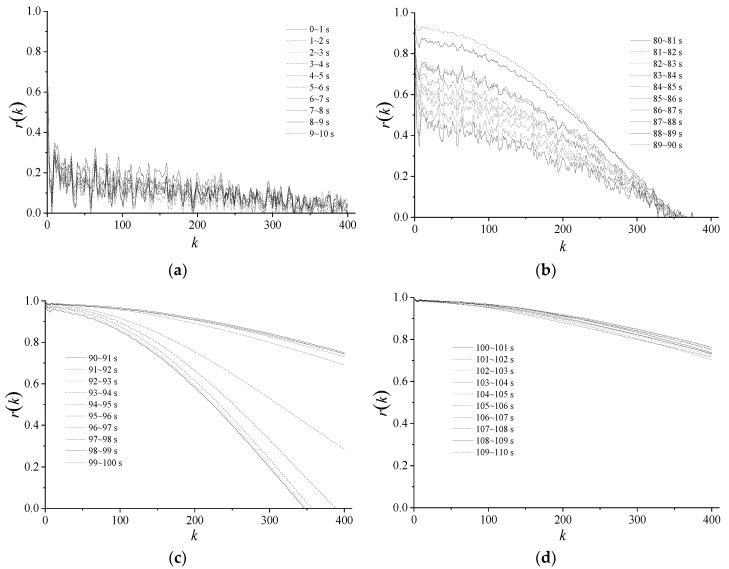
Autocorrelation function of dynamic pressure. (**a**) 0–10 s; (**b**) 80–90 s; (**c**) 90–100 s; (**d**) 100–110 s.

**Figure 5 entropy-21-00266-f005:**
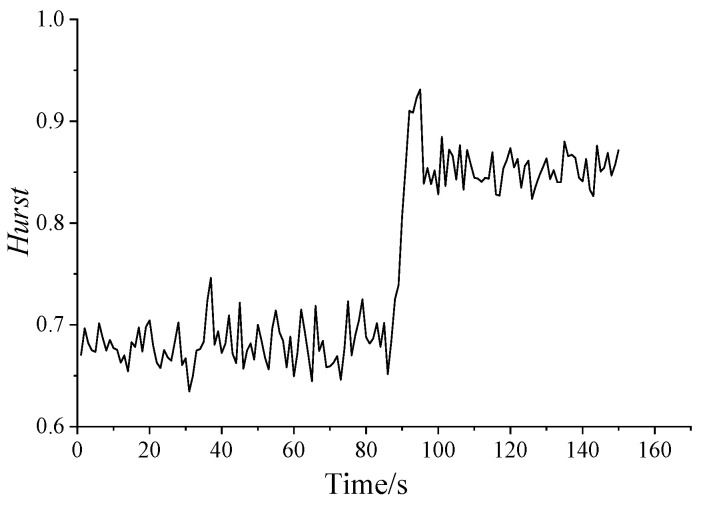
Hurst exponent diagram.

**Figure 6 entropy-21-00266-f006:**
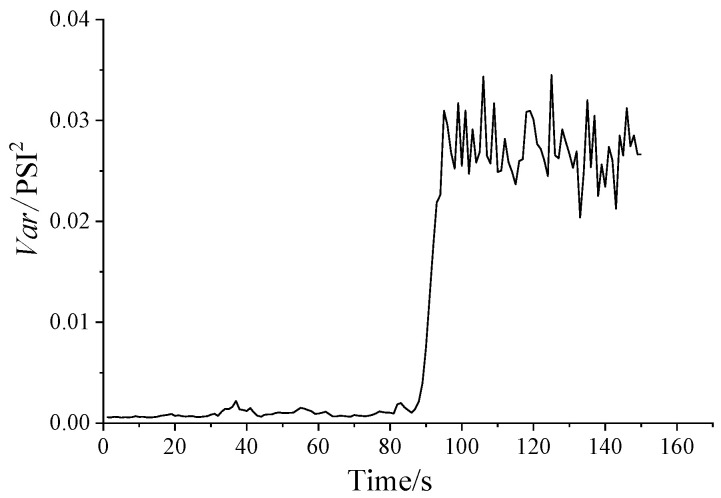
The variance diagram.

**Figure 7 entropy-21-00266-f007:**
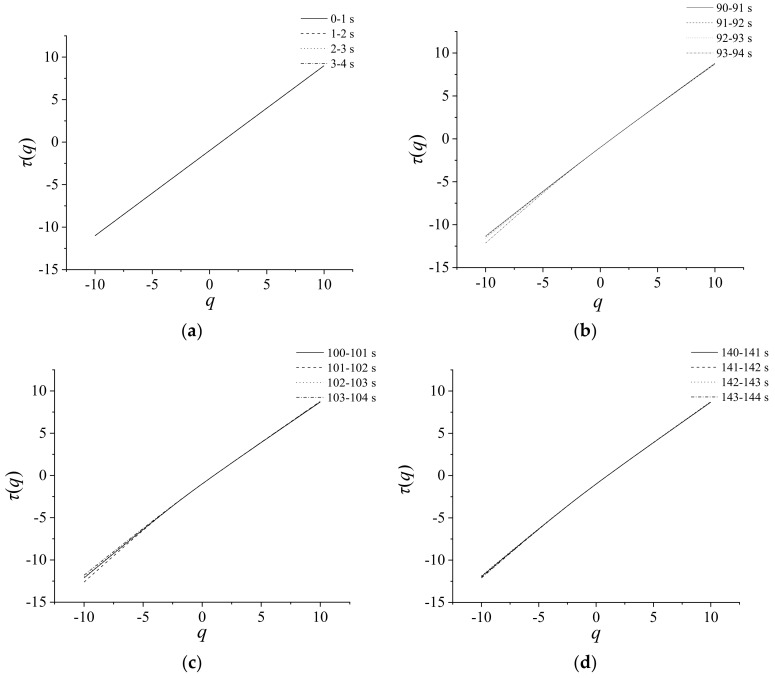
Structure function of dynamic pressure. (**a**) 0–4 s (**b**) 90–94 s; (**c**) 100–104 s; (**d**) 140–144 s.

**Figure 8 entropy-21-00266-f008:**
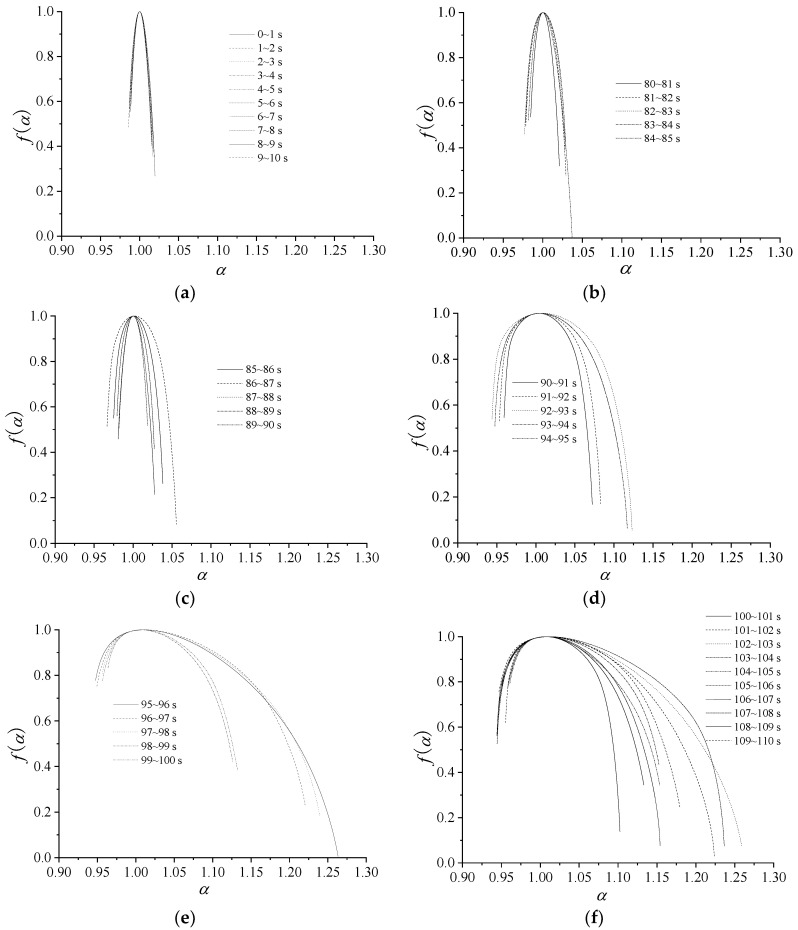
Multi-fractal spectrum of dynamic pressure. (**a**) 0–10 s; (**b**) 80–85 s; (**c**) 85–90 s; (**d**) 90–95 s; (**e**) 95–100 s; (**f**) 100–110 s.

**Figure 9 entropy-21-00266-f009:**
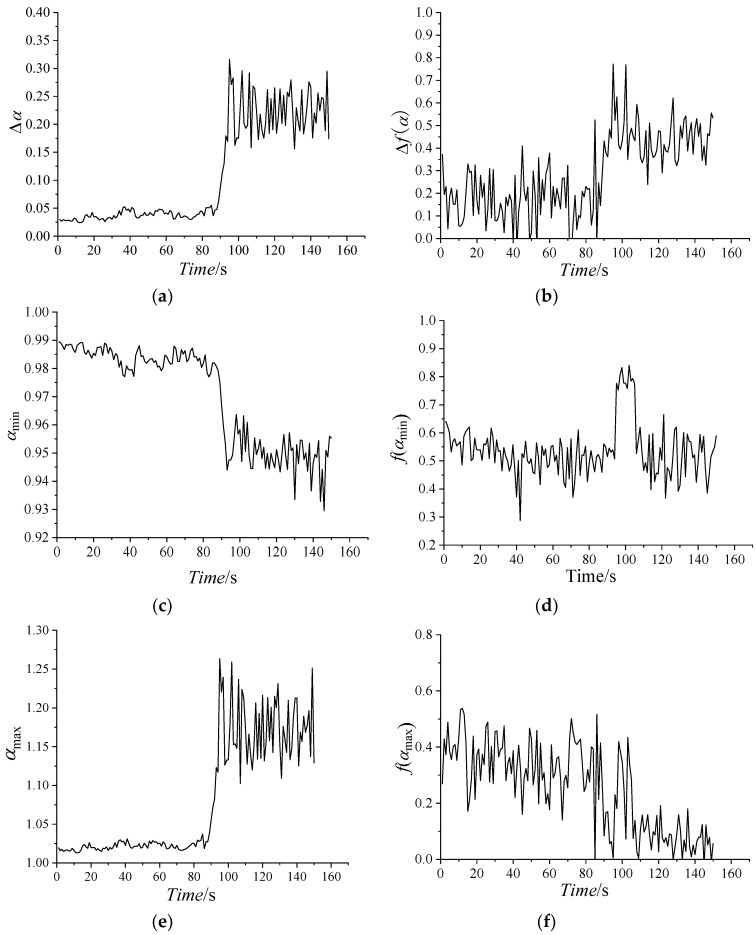
Variation of multi-fractal spectrum parameters with time. (**a**) Δ*α*; (**b**) Δ*f*(*α*); (**c**) *α_min_*; (**d**) *f*(*α_min_*); (**e**) *α_max_*; (**f**) *f*(*α_max_*).

**Figure 10 entropy-21-00266-f010:**
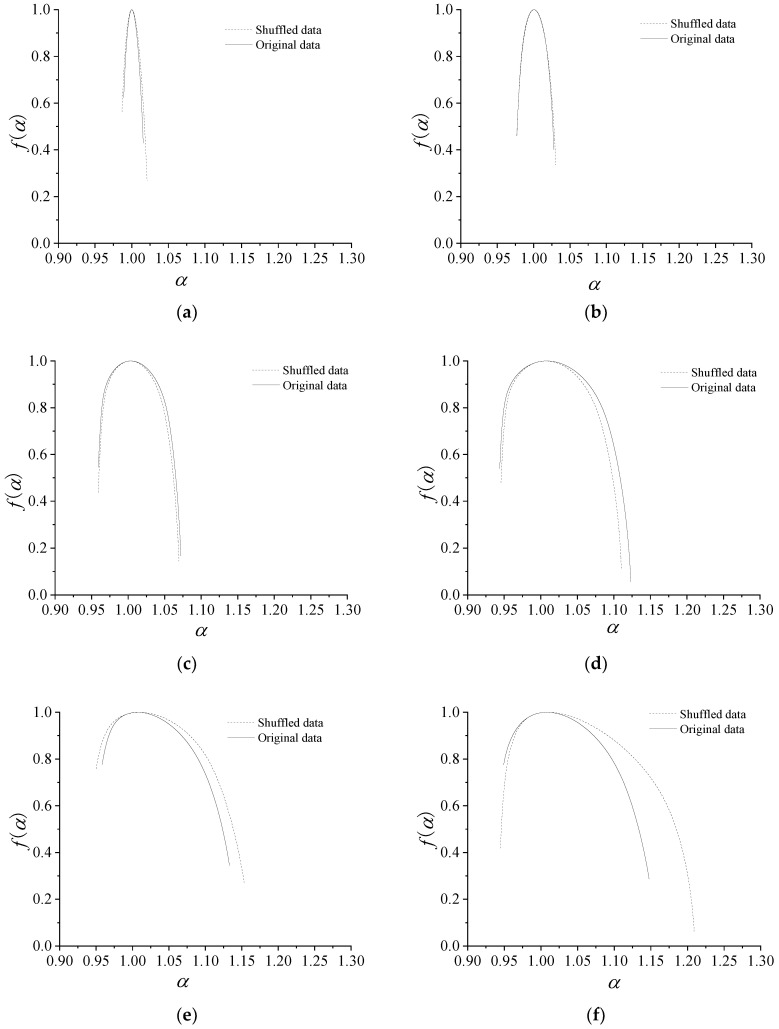
Multi-fractal spectrum parameters of the original and randomly shuffled dynamic pressure. (**a**) 1–2 s; (**b**) 82–83 s; (**c**) 90–91 s; (**d)** 92–93 s; (**e**) 100–101 s; (**f**) 105–106 s.

**Figure 11 entropy-21-00266-f011:**
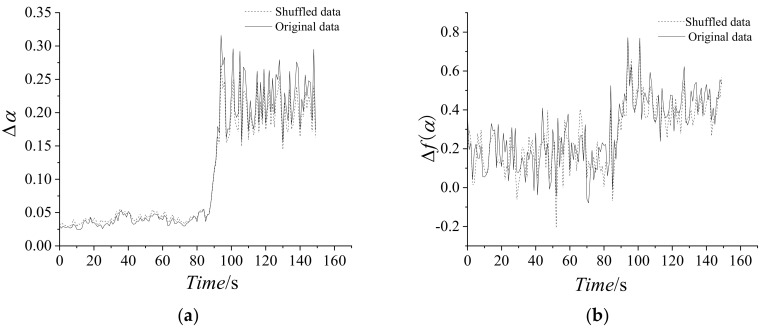
Variation of multi-fractal spectrum parameters with time for the randomly shuffled dynamic. (**a**) Δ*α*; (**b**) Δ*f*(*α*).

**Table 1 entropy-21-00266-t001:** Acquisition system parameters.

Driving Motor Power	Diffuser Blade Number	Blade Number	Rotor Velocity	Mach Number	Dynamic Acquisition System	Sample Frequency
800 kW	24	16	960 r/min	0.6	CoCo80	20.48 kHz

**Table 2 entropy-21-00266-t002:** Hurst exponent and variance in 81–100 s for dynamic pressure.

**Time/s**	81	82	83	84	85	86	87	88	89	90
**Hurst**	0.6816	0.6867	0.7018	0.6783	0.7019	0.6515	0.6841	0.7253	0.7391	0.8069
**Variance**	9.50E-04	0.0019	0.002	0.0016	0.0013	0.001	0.0014	0.0022	0.004	0.0075
**Time/s**	91	92	93	94	95	96	97	98	99	100
**Hurst**	0.8578	0.9102	0.9083	0.9231	0.9312	0.8385	0.854	0.8381	0.8518	0.8281
**Variance**	0.0125	0.0175	0.0219	0.0226	0.031	0.0295	0.027	0.0252	0.0317	0.0255

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
