# Peer review of "Descriptions of Entropy with Fractal Dynamics and Their Applications to the Flow Pressure of Centrifugal Compressor"

_entropy, 2019, doi:10.3390/e21030266_

Round 1
Reviewer 1 Report
I read the new version of the manuscript "Descriptions of Entropy with Fractal Dynamics and Their Applications to the Flow Pressure of Centrifugal Compressor", by Yan Liu, Dongxiao Ding, Kai Ma, and Jiazhong Zhang. I am of the same opinion that, while the authors analyze the pressure fluctuations of a centrifugal compressor, the method does not have predicting power, as the authors claim. Therefore, if the paper will be published, in my opinion, this claim should be removed.
The paper is not well written and is quite hard to read. I give you some examples, from the situations which are easier to correct:
- Between the value and the unit of a quantity there should be a space. For example, in the abstract, "800kw" should be replaced by "800 kW".
- There are awkward constructions, like "As a conclusion, it can be concluded ...", that appears in the abstract.
- On page 2, line 89, appears "lag k". What is this?
- "tails are not exponentially bounded" should be reformulated.
- In an article, equations are part of phrases. On page 3, lines from 108 to 119 are not part of any phrase. Please reformulate.
etc.
It is hard for me to make any recommendation. I suppose that the data and the analysis would deserve publication, but it is hard to follow and the presentation quality is far from the readers' expectations. Overall, in my opinion, the paper does not meet the quality of the journal.
Author Response
Dear Editor and Reviewers,
Thanks very much for reviewing and the kind suggestions. The followings are the list of the improvements of my manuscript.
The paper is not well written and is quite hard to read. I give you some examples, from the situations which are easier to correct:
- Between the value and the unit of a quantity there should be a space. For example, in the abstract, "800kw" should be replaced by "800 kW".
Response: The sentence has been revised, so do the other sentences.
- There are awkward constructions, like "As a conclusion, it can be concluded ...", that appears in the abstract.
Response:The sentence is revised as “As a conclusion, the proposed method, as one measure method for entropy, can be used to feasibly identify the incipient surge of a centrifugal compressor and design its surge controller.”
- On page 2, line 89, appears "lag k". What is this?
Response: The sentence has been moved to the later of Eq. (1), and is revised as “where k is time lag. For a stationary process, the auto-correlation function attenuates to zero rapidly along with the increase of k.”
- "tails are not exponentially bounded" should be reformulated.
Response: The sentence is revised as “In probability theory, heavy-tailed distributions are probability distributions which have heavier tails than the exponential distribution [14].”
- In an article, equations are part of phrases. On page 3, lines from 108 to 119 are not part of any phrase. Please reformulate.
Response: These lines has been reformulated.

Reviewer 2 Report
The authors improved significantly the quality of the paper according to some of my suggestions. However, I still have one critical remark. I strongly encourage the authors to perform a test of statistical reliability of the MF results as I indicated in my previous review (point 9). The authors should perform MF analysis of randomly shuffled data and/or data for which Fourier phases in frequency domain are randomly shuffled and then compare the results between obtained for surrogates and original data (see similar analysis in ref [21]).
Author Response
Dear Editor and Reviewers,
Thanks very much for reviewing and the kind suggestions. The followings are the list of the improvements of my manuscript.
1. The authors improved significantly the quality of the paper according to some of my suggestions. However, I still have one critical remark. I strongly encourage the authors to perform a test of statistical reliability of the MF results as I indicated in my previous review (point 9). The authors should perform MF analysis of randomly shuffled data and/or data for which Fourier phases in frequency domain are randomly shuffled and then compare the results between obtained for surrogates and original data (see similar analysis in ref [21]).
Response: section 7 “ The statistical reliability of multi-fractal spectrum for dynamic pressure” has been added to the manuscript to discuss the statistical reliability of the MF results.